# Fatty Acid-Binding Protein 4-Mediated Regulation Is Pivotally Involved in Retinal Pathophysiology: A Review

**DOI:** 10.3390/ijms25147717

**Published:** 2024-07-14

**Authors:** Hiroshi Ohguro, Megumi Watanabe, Fumihito Hikage, Tatsuya Sato, Nami Nishikiori, Araya Umetsu, Megumi Higashide, Toshifumi Ogawa, Masato Furuhashi

**Affiliations:** 1Departments of Ophthalmology, School of Medicine, Sapporo Medical University, S1W17, Chuo-ku, Sapporo 060-8556, Japan; watanabe@sapmed.ac.jp (M.W.); fuhika@gmail.com (F.H.); nami076@yahoo.co.jp (N.N.); araya.umetsu@sapmed.ac.jp (A.U.); megumi.h@sapmed.ac.jp (M.H.); 2Departments of Cardiovascular, Renal and Metabolic Medicine, Sapporo Medical University, S1W17, Chuo-ku, Sapporo 060-8556, Japan; satatsu.bear@gmail.com (T.S.); a08m024@yahoo.co.jp (T.O.); 3Departments of Cellular Physiology and Signal Transduction, Sapporo Medical University, S1W17, Chuo-ku, Sapporo 060-8556, Japan

**Keywords:** fatty acid-binding protein 4 (FABP4), fatty acid-binding protein 5 (FABP5), vascular endothelial growth factor A (VEGFA), retinal vascular disease (RVD), proliferative diabetic retinopathy (PDR), retinal vein occlusion (RVO)

## Abstract

Fatty acid-binding proteins (FABPs), a family of lipid chaperone molecules that are involved in intracellular lipid transportation to specific cellular compartments, stimulate lipid-associated responses such as biological signaling, membrane synthesis, transcriptional regulation, and lipid synthesis. Previous studies have shown that FABP4, a member of this family of proteins that are expressed in adipocytes and macrophages, plays pivotal roles in the pathogenesis of various cardiovascular and metabolic diseases, including diabetes mellitus (DM) and hypertension (HT). Since significant increases in the serum levels of FABP4 were detected in those patients, FABP4 has been identified as a crucial biomarker for these systemic diseases. In addition, in the field of ophthalmology, our group found that intraocular levels of FABP4 (ioFABP4) and free fatty acids (ioFFA) were substantially elevated in patients with retinal vascular diseases (RVDs) including proliferative diabetic retinopathy (PDR) and retinal vein occlusion (RVO), for which DM and HT are also recognized as significant risk factors. Recent studies have also revealed that ioFABP4 plays important roles in both retinal physiology and pathogenesis, and the results of these studies have suggested potential molecular targets for retinal diseases that might lead to future new therapeutic strategies.

## 1. Introduction

Fatty acid-binding proteins (FABPs) are structurally conservative water-soluble cytoplasmic proteins with a relatively small molecular weight of around 15,000 Da. They are involved in various lipid-related biological functions including oxidation, signaling, regulation of gene transcription, and storage by transporting fatty acids (FAs) to intracellular compartments [1,2,3,4]. So far, 12 FABP family members have been identified and two of them (FABP 10 and FABP 11) are not expressed in humans [5]. The spatial structure of these FABPs share a common so-called β-barrel structure composed of a central hydrophobic core surrounding 10 antiparallel chains in two vertical directions [6], and diversity in protein sequences contributes to the distinguishability of binding ligands to induce specific biological ability among the individual members [1,2,3,4]. In addition to the physiological roles of FABPs, numerous studies have shown that FABPs are also involved in the development and progression of the pathogenesis of various diseases, including cardiovascular, renal, endocrine and metabolic diseases, cancers, and neurodegenerative diseases [2,3,4]. As members of the human FABP family of proteins, the following ten isoforms have been identified: liver (L-FABP/FABP1), intestinal (I-FABP/FABP2), heart (H-FABP/FABP3), adipocyte (A-FABP/FABP4/aP2), epidermal (E-FABP/FAPB5/mal1), ileal (Il-FABP/FABP6), brain (B-FABP/FABP7), myelin (M-FABP/FABP8), testis (T-FABP/FABP9) [2,3,4], and FABP12 [7] (Table 1). As for ocular FABPs, a recent single-cell transcriptomic investigation showed that several FABPs are expressed among various cell components within the human retina [8], suggesting that FABPs may also have significant roles in the physiology, as well as the pathology, of the retina. Furthermore, our research group revealed that FABP4 is a main subspecies of intraocular FABPs (ioFABPs) and is involved in both the physiology and pathogenesis of intraocular tissues, especially the retina [9,10,11,12]. Therefore, in this review, we present recent findings, with a focus on the biological significance of ioFABPs, especially ioFABP4, as well as their substrates, intraocular FFAs (ioFAs), and their contribution to retinal pathophysiology.

## 2. Intraocular Free Fatty Acids (ioFFAs) 

Physiologically, FFAs are essential elements for almost all living organisms because they serve not only as a source of energy but also as a component of all cellular membranes in the form of phospholipids [22]. In vertebrates, including humans, redundant FFAs are stored in the form of triglycerides and cholesterol esters in adipocytes and intracellular lipid droplets and can be used on demand to maintain lipid-related homeostasis [22]. On the other hand, excess FFAs are also involved in several pathological states such as obesity, DM, fatty liver, and dyslipidemia [23,24]. Thus, FFAs need to be tightly regulated at both the intracellular and extracellular levels.

FAs are generally ingested in the form of phospholipids and triglycerides, and in turn, Triglycerides are hydrolyzed into mono- and diglycerides and FFAs during digestion [25,26]. This type of FA metabolism occurs in most cells and is particularly important in adipocytes, in which FAs are re-esterified to form triglycerides that are then stored in the form of fat droplets [27]. However, when triglycerides in chylomicrons are cleaved, the resulting FAs are not always directly taken up by nearby cells and some are transported through the circulation to other cells as forms that are bound to serum albumin [28]. Alternatively, another source of plasma FAs is the so-called de novo lipogenesis, that is, their endogenous synthesis from carbohydrates [26,27]. This de novo lipogenesis can also occur in most cells, particularly in the liver and adipose tissue, as well as in mammary glands. After the FAs are absorbed into cells or undergo de novo lipogenesis, most of the FAs are then transported by intracellular FABPs to mitochondria, where they undergo β-oxidation to produce energy in the form of ATP [27,29]. On the other hand, FAs can also be converted into phospholipids or sphingolipids, which function as major components of cellular membranes and have important intracellular signaling roles [30,31,32,33,34,35]. Among the FA subspecies, it is shown that saturated FAs (SFAs) have a clearly different origin, metabolism, and function than those of polyunsaturated fatty acids (PUFAs). In fact, brain SFAs, such as palmitate (16:0) or stearic (18:0), are known to be generated in situ, as well as being imported into the brain, whereas PUFAs, which are elongated and further unsaturated mainly in the liver, are then transported through the blood circulation and imported into neural tissue as non-esterified FAs [36,37,38]. Thus, both endogenous and blood-derived PUFAs accumulate preferentially in neurons in the form of phospholipids. 

### 2.1. Contribution of ioFFA to Retinal Pathogenesis

Within intraocular tissues, it has been shown that photoreceptor outer segments contain abundant lipids including phospholipids (90–95%) and cholesterols (4–6%) [39,40], although there are no evident fatty tissues and cells. Furthermore, it is known that the human retina contains five major types of FFAs: palmitic acid (C16:0), stearic acid (C18:0), oleic acid (C18:1), linoleic acid (C18:2), and arachidonic acid (C20:4) [41]. Those studies suggest that lipid metabolisms, especially FFAs, may be an important regulatory mechanism of intraocular physiology. In our recent study, we detected SFAs (C16:0 and C18:0), monounsaturated fatty acids (MUFA) (C18:1), and PUFAs (C18:2 and C20:4) as major vitreous FA subspecies of ioFFAs in vitreous specimens obtained from patients with retinal vascular diseases (RVDs) and non-RVDs [12]. Interestingly, the levels of ioFFAs and intraocular FABP4 (ioFABP4) were significantly, but independently, elevated in patients with RVDs compared with levels of non-RVD patients [12]. As of this writing, the origins of ioFFAs have not been elucidated yet. However, no significant correlation between plasma (p FFAs and ioFFAs (Table 2) in our recent study suggested that ioFFAs may be generated locally in addition to leaking from the bloodstream during the pathogenesis of RVD. In fact, it has been reported that an increase in vascular permeability and breakdown of the blood-retinal barrier (BRB) occur during the progression of these diseases [42]. Furthermore, FABP4, which is primarily regarded as an adipocyte- and macrophage-specific protein [2,16,17], is also expressed in capillaries and veins [43], and our recent study also showed that the levels of ioFABP4 and ioFFAs were closely correlated, either with or without RVD. 

It has been shown that lipids account for approximately one-third of the retina dry weight, and 87%, 11%, and 1.7% of the total lipids in the human retina are phospholipid, free cholesterol, and cholesterol esters, respectively [40]. The retinal pigment epithelium (RPE) and ocular choroid also contain many lipids, including 58% phospholipid, 19% cholesterol ester, 15% free cholesterol, 4% triglycerides, and 4% FFAs. DHA, stearic acid, and palmitic acid have been identified as the most prevalent FFA subspecies in the retina [40,44]. Despite such abundant FFAs within the intraocular environment, the biological roles of the FFAs have not been elucidated, although some of those species, including DHA, have been extensively studied [45,46]. For instance, it was shown that DHA in disc membranes of the photoreceptor outer segment greatly influences the phototransduction pathway by optimizing the conformational states of rhodopsin molecules during light absorption. Therefore, DHA in the photoreceptor disc membrane may precisely maintain the FFA compositions in the retina so that they are not influenced by dietary fluctuations [47,48,49]. However, in the case of a decrease in retinal DHA levels following chronic DHA deficiency, there are functional deficits in an electroretinogram (ERG) [49] and loss of vision [48], presumably due to morphological changes of the photoreceptor outer segment disc membranes [50]. 

### 2.2. Possible Effects of ioFFA on the Outer Blood-Retinal Barrier (oBRB)

In addition to photoreceptor cells, it has been shown that RPE cells maintain lipid metabolism homeostasis in the retina through the use of lipid metabolism-related enzymes, with mitochondrial fatty acid β-oxidation being a major pathway for fatty acid degradation [51], and lipid metabolism regulation in the RPE has therefore been suggested to be a possible therapeutic strategy for various retinal diseases such as age-related macular degeneration (AMD) [52]. However, among these lipid metabolism regulations in the RPE, the effects of ioFFAs have not yet been elucidated. It is known that the oBRB is composed of the ocular choroid, Bruch’s membrane (BM), and RPE [53], and rigid intercellular binding by tight junctions (TJs) in the RPE monolayer sheet of the oBRB mainly functions as a biological barrier to regulate trafficking of nutrients, wastes, and inflammatory cells between the ocular choroid and outer retina [54,55,56,57]. In addition, TJs of the RPE are located at the apical surface and thus contribute to the maintenance of interepithelial junctional integrity and permeability [58,59,60,61]. A previous study using human intestinal epithelial Caco-2 cells showed that heat stress after pre-incubation with EPA, DHA, or arachidonic acid effectively attenuated the decrease in trans-epithelial electric resistance (TEER) and significantly increased the expression levels of occludin and ZO-1 [62], and another study showed that DHA- and EPA-enriched phosphatidylcholine enhanced the permeability across monolayers of Caco-2 cells [63], suggesting that ioFFA may also affect oBRB functions.

## 3. FABP4 Is a Main Isoform of the FABP Family, among ioFABPs 

Sellner and Phillips first showed the fatty acid preferences of ocular FABP isoforms and the effects of the FABP isoforms by measuring the acylation of 1-palmitoyl-sn-glycerophosphocholine (1-16:0-GPC) or 1-palmitoyl-sn-glycerophosphoethanolamine (1-16:0-GPE) using microsomal fractions prepared from the retinas of 14–15-day-old chick embryos [64]. Thereafter, Sellner showed by immunohistochemistry that a retinal FABP homologous to mammalian H-FABP (FABP3) is localized in photoreceptor inner segments, the outer nuclear layer (ONL), the ganglion cell layer (GCL), and the inner limiting membrane in the embryonic chick retina [65,66]. Therefore, they suggested that this retinal FABP may play an important role in FA-related regulations during the development of the retina. Another study using in situ hybridization showed the presence of FABP12 in the retinal ganglion and inner nuclear layer cells in addition to the testis of rats and in human retinoblastoma cell lines [7]. In another immunohistochemical study, immunolabeling was detected in amacrine/bipolar/horizontal interneurons, microglia, ganglion cells, and cone photoreceptor cells by anti-H-FABP (FABP3), anti-A-FABP (FABP), anti-E-FABP (FABP5), and anti-B-FABP (FABP7), respectively, in a healthy mouse retina [67]. Interestingly, immunoreactivity by anti-E-FABP (FABP5) was recognized in invasive macrophages in a photopic-damaged mouse retina [67], suggesting that E-FABP (FABP5) may be involved in retinal pathogenesis. In support of these results, a recent study using a single-cell transcriptomic analysis also showed that several FABP isoforms, including FABP3, FABP4, FABP5, FABP7, FABP8, and FABP12, are diversely expressed in human retinal cells [8]. Collectively, these findings suggest that various FABP isoforms are indeed present within vertebrate retinas and may be differently involved in some biological roles in the retina. To study this issue further, our group independently performed immunohistochemistry using healthy human retinas and rodent retinas from wild-type (WT) and disease models of diabetic retinopathy (DR) or retinitis pigmentosa (RP) by using antibodies against FABP3, FABP4, FABP5, FABP7, FABP8, and FABP12 [12]. As shown in Table 3, immunohistochemistry revealed that positive labeling profiles in the healthy human retinas were exclusively different among FABP isoforms. All retinal layers, those except the photoreceptor outer segment (OS), or those except the OS and RPE were immunolabelled by anti-FABP3 antibody, anti-FABP4 antibody (Figure 1), or anti-FABP7 antibody, respectively, and anti-FABP8 antibody reacted with the nerve fiber layer (NFL), inner plexiform layer (IPL) and RPE. On the other hand, in rodent retinas, only immunoreactivities against FABP4 and FABP12 were detected, and retinal immunolabeling patterns by anti-FABP4 were different among WT, DR, and RP rodent retinas. Therefore, these collective data suggest that FABP4 is a major retinal FABP isoform and FABP4 may have a pivotal role in retinal pathophysiology.

## 4. Roles of ioFABP4 in Retinal Pathogenesis

FABP4 was initially known as an adipocyte-derived protein that functionally plays an important role in the maintenance of glucose and lipid homeostasis [1,68]. However, recent observations have suggested that FABP4 is more widely expressed than initially thought, and FABP4 has been shown to be expressed in capillary and venous cells, rather than in arterial, endothelial cells under physiological conditions [2,17]. In addition, among FABP proteins, FABP4 is also expressed in macrophages, in addition to adipocytes, and plays a pivotal role in the pathogenesis of cardiovascular diseases, HT, DM, and cancer [69,70,71,72,73,74]. Furthermore, FABP4 can be secreted into various bodily fluids such as plasma and, in fact, significant increases in the serum levels of FABP4 have been detected in patients with cardiovascular and metabolic diseases [1,68,75,76,77,78,79,80,81,82]. Since several of these diseases related to FABP4 are also well-known as key risk factors for various RVDs including proliferative DR (PDR), retinal vein occlusion (RVO), and AMD, it has been rationally suggested that FABP4 is involved in the pathogenesis of RVDs. 

### 4.1. Proliferative Diabetic Retinopathy (PDR)

PDR is known as an advanced stage of DR with serious retinal complications, which can induce vision loss among relatively young DR patients worldwide [83]. As the underlying mechanism for the pathogenesis of PDR, it has been shown that numerous biochemical and inflammatory processes in response to long-term exposure to hyperglycemia simultaneously induce vascular endothelial dysfunction, pericyte loss, and neurodegeneration, which lead to the development of hypoxia and neovascularization [84]. During the course of disease progression, the expression of various cytokines including vascular endothelial growth factor (VEGF), tumor necrosis factor-alpha (TNF-α), and inducible nitric oxide synthase is locally induced in the diabetic retina in response to hypoxia [85]. In addition to the accumulation of chemokines, adhesion molecules such as intercellular adhesion molecule-1 are also induced to facilitate the migration of leukocytes into the retinal endothelium, and thereby vascular permeability increases and the inner blood-retinal barrier is broken down [42]. Therefore, this VEGF-related signaling is known as the main mechanism of the pathogenesis of PDR and provides a rationale therapeutic strategy to use anti-VEGF drugs [86] in addition to traditional anti-angiogenic agents [87], intravitreal injections of corticosteroids, and/or laser photocoagulation therapy [88]. Nevertheless, these single or combined therapies have only provided limited success in the treatment of PDR [86,87,88], and additional target molecules other than VEGF are required for therapy for PDR. 

In our recent study, we focused on ioFABP4 as an additional candidate of a pathogenic target molecule for PDR and examined the concentrations of ioFABP4 and intraocular VEGFA (ioVEGFA) in vitreous specimens surgically obtained from patients with PDR (n = 20) [10,11,12]. As non-PDR controls (n = 20), vitreous specimens were also collected from non-DR patients with epiretinal membrane (ERM). As expected, both vitreous concentrations of ioFABP4 and ioVEGFA determined by enzyme-linked immunosorbent assays were substantially increased in eyes with PDR, and a strong positive correlation (r = 0.72, *p* < 0.001) was observed between levels of ioFABP4 and ioVEGFA (Table 4 and Table 5). Furthermore, both factors were negatively correlated with ocular blood flow in the optic nerve head, and the correlation was stronger for ioFABP4 (Table 6). However, correlation analyses with various clinical factors and stepwise multiple regression analyses suggested that ioFABP4 and ioVEGFA were independently regulated. Levels of ioFABP4 were not correlated with the presence of vitreous hemorrhaging or plasma levels [10,11,12]. Collectively, the results suggested that ioFABP4 originates from some intraocularly originating cells, not from peripheral blood circulation, and that ioFABP4 may be an additional key target molecule involved in the pathogenesis of PDR by affecting ocular blood circulation. 

Regarding relationships between FABP4 and VEGFA, previous studies showed that VEGFA and basic fibroblast growth factor (bFGF) can facilitate the expression of FABP4 in endothelial cells, and FABP4 thereby stimulates angiogenesis [89]. VEGFA-induced FABP4 expression was inhibited by knockdown of VEGF receptor-2, and knockdown of FABP4 substantially reduced the proliferation of endothelial cells, regardless of the absence or presence of stimulation by VEGF and bFGF [43]. It was also shown that the expression of FABP4, but not that of VEGFA, is induced by cellular senescence, oxidative stress, and injury in microvascular endothelial cells [90] or arterial endothelial cells [91], supporting the above idea that FABP4 may be related to ocular blood circulation. 

It has also been shown that FABP4 levels are significantly influenced by several chemicals and drugs, including a statin [92], eicosatetraenoic acid (EPA)/docosahexaenoic acid (DHA) agent [93], dipeptidyl peptidase 4 inhibitor [94], and angiotensin II receptor blocker (ARB) [95]. Interestingly, it has been reported that angiotensin II and components of the renin-angiotensin system (RAS) are expressed in the retina [96]. In fact, it is thought that angiotensin II stimulates retinal leukostasis by activation of the angiotensin type 1 receptor signaling pathway, thereby stimulating the production of proinflammatory and proliferative mediators, resulting in the development and progression of PDR [97], as well as choroidal neovascularization (CNV) [98]. Furthermore, selective angiotensin receptor blockers have been shown to have neuroprotective and anti-inflammatory effects on retinal angiogenesis and neovascularization in animal models [99,100,101]. Based on these findings, it was shown in several clinical trials that inhibiting the RAS by an ARB reduced the incidence and progression of DR [102]. Therefore, the relationship between ARB and FABP4 could provide additional proof that FABP4 indeed plays a pivotal role in the pathogenesis of DR, and FABP4 may be a promising therapeutic key molecule for the treatment of DR progression. In support of these observations, a recent study using a mouse model of streptozocin (STZ)-induced DR and high glucose-treated adult retinal pigment epithelium 19 (ARPE-19) cells showed that inhibition of FABP4 by BMS309403 alleviates lipid peroxidation and oxidative stress in DR by regulating peroxisome proliferator-activator receptor-mediated ferroptosis [103].

### 4.2. Retinal Vascular Occlusion (RVO)

RVO is recognized as a common RVD and is clinically categorized into central retinal vein occlusion (CRVO), hemi-central vein occlusion (hemi-CRVO), and branch retinal vein occlusion (BRVO). The former and the latter are caused by thrombosis at the lamina cribrosa, and the second occurs at an intersection of a branched retinal artery and vein, respectively [104,105,106,107,108]. As in the case of PDR, RVO is often associated with visual deterioration induced by retinal edema, ischemia, and neovascularization [109], in which VEGF is known to contribute as the possible underlying mechanism [110]. Therefore, intravitreous anti-VEGF therapy is currently used as the main therapy for vision-threatening patients with RVO [111,112,113]. However, although anti-VEGF therapy has remarkable therapeutic effects, its effects are usually transient and thus this mono-treatment cannot stop the progression of RVO [114,115,116,117], and thus, additional therapeutic target molecules, other than VEGF, urgently need to be identified. 

As in the case of PDR, our group speculated that FABP4 may also be a key target molecule in the case of RVO and we therefore examined the levels of ioFABP4 and ioVEGFA using vitreous specimens surgically obtained from patients with RVO or ERM [9]. As expected, we found significantly increased levels of ioFABP4 and ioVEGFA in patients with RVO compared with levels in patients with ERM (*p* < 0.05) [9] and a significantly positive correlation (r = 0.36, *p* = 0.045) [9] between ioFABP4 and ioVEGFA (Table 4 and Table 5). Furthermore, several correlation analyses showed that both ioFABP4 and ioVEGFA were also independently regulated in RVO, as in PDR [10]. Therefore, FABP4 may be involved in the pathogenesis of both RVO and PDR as an independent key factor, in addition to VEGFA. 

### 4.3. Age-Related Macular Degeneration (AMD)

AMD has been recognized as the most common cause of permanent visual deterioration in the aged population worldwide, resulting in serious problems in public health [118,119]. As the main pathogenic process of AMD, choroidal neovascularization (CNV) is known to lead to severe vision loss in neovascular AMD due to increased expression of ioVEGF [120,121], and anti-VEGF therapies have been successfully used to treat neovascular AMD [122]. However, such anti-VEGF therapies have several limitations, including the requirement of repeat therapy, development of drug resistance, and large costs for patients, but, unfortunately, there is currently no alternative and satisfactory treatment for neovascular AMD [123]. Therefore, other options for cost-effective, less invasive, and more durable therapy for CNV in AMD patients are required. For this purpose, a therapeutic target molecule that substitutes for VEGF will be needed. As of this writing, there is no evidence of a contribution to FABP4 on AMD pathogenesis. However, previous studies have shown a significant contribution of macrophages to CNV formation in animal models and AMD patients [124,125,126], and levels of cytokine production and proinflammatory mediators, including TNFα and COX2, were reduced in macrophages isolated from fabp4-deficient (*fabp4*^−/−^) mice [127], suggesting that FABP4 may be involved in the pathogenesis of AMD. To support this possibility, a previous study using an oxygen-induced retinopathy (OIR) model [128] in wild-type (WT) and *fabp4*^−/−^ mice showed that OIR induction in *fabp4*^−/−^ mice caused a significant decrease in neovessel formation and a significant improvement in physiological revascularization of avascular retinal tissues [129]. Furthermore, it has been shown that FABP4 is an inducible factor for angiogenesis and vascular smooth muscle cell proliferation and migration, and FABP4 has therefore been established as a reliable predictive biomarker for cardiovascular disease in specific at-risk groups [130]. Collectively, the observations suggested that ioFABP4 may be a new risk factor, as well as a possible therapeutic target for CNV in AMD. 

## 5. Roles of Intraocular FABP4 (ioFABP4) in Retinal Physiology

In addition to the evidence of physiological expression of FABP4 in capillary venous and endothelial cells [2,17], our recent studies showed that ioFABP4 was present in vitreous specimens, even from patients without RVDs [9,10,12], suggesting that ioFABP4 may have some physiological roles. In a recent study, to elucidate the unknown physiological roles of ioFABP4, electroretinograms (ERGs) of WT and *fabp4*^−/−^ mice were analyzed and it was found that both ERG a- and b-waves were larger in *fabp4*^−/−^ mice than in WT mice (Figure 2) [12], suggesting that ioFABP4 may be involved in the crucial regulatory mechanism of the retinal phototransduction pathway. To obtain additional insight into the physiological significance of ioFABP4, its intraocular origin was investigated using four representative intraocular tissue-derived cell types including human non-pigmented ciliary epithelium cells, retinoblastoma cells, ARPE19 cells, and human ocular choroidal fibroblast (HOCF) cells [131]. Based on the result that, among these cells, gene expression of FABP4 was only detected in HOCF cells, we suggested that ocular choroidal tissue is one of the possible producing cells of ioFABP4 within the intraocular environment. However, since gene expression in these immortalized cell lines may not reflect in vivo expression of genes in their corresponding cell types in the retina, additional experimental evidence will be required. 

The ocular choroid is in physical contact with the RPE and supplies various nutrients, oxygen, and biological factors via blood circulation to the sensory outer retina [132]. The RPE, a monolayer of polarized epithelial cells, is also in direct contact with OS to maintain photoreceptor survival and functions related to phototransduction by daily phagocytosis of the top of OS tips and recycling of11-cis retinal, in addition to its biological role as the BRB [133]. Since we detected that positive immunoreactivities against FABP4 were detected in all of the retinal segments except the OS [12], we reasonably speculated that ioFABP4 originated from the ocular choroid and spread toward the sensory retina and vitreous cavity beyond the RPE. If this speculation is correct, ioFABP4 would affect not only the initial phototransduction by the OS but also the following phototransduction by the mid-retina, which are the origins of the a-wave and b-wave of an ERG, respectively, as observed in the ERG in *fabp^−/−^* mice described above (Figure 2), suggesting that the ocular choroid may form a putative complex where the ocular choroid and the sensory retina function together in ocular pathophysiology. In fact, various studies have already advocated the concept of an RPE/ocular choroid complex [134,135], and that complex has been suggested to be involved in various ocular diseases including AMD [136] and myopia [137]. More interestingly, we also found by Seahorse extracellular flux analysis that inhibition of FABP4 by BMS309403 induced so-called pseudohypoxic states (Figure 3). Such pseudohypoxia states may lead to neovascularization, as was suggested by using an A”D model [138], and malignant tumors [139,140], and this scenario may strongly support our observations, indicating that FABP4 is an independent key pathogenic factor for RVDs [12] such as DR [10] and RVO [9] in addition to intraocular physiology. Previous studies also showed that FABPs are required for development of the retina and BRB in zebrafish [141,142] and chickens [143,144], and another study using a Drosophila ninaEG69D mutant showed that FABP is required for light-induced Rh1 degradation and photoreceptor survival [145]. These collective findings suggest that ioFABP4 originates from the ocular choroid and may be a critical regulator for cellular homeostasis of non-adipocyte HOCF cells, thereby importantly contributing to ocular pathophysiology.

## 6. Fatty Acid-Binding Protein 5 (FABP5)

It has been shown that FABP5 is an important coordinator of intracellular FAs and that it functions to increase the solubility of FAs and reversibly binds with high-affinity hydrophobic ligands such as saturated and unsaturated long-chain fatty acids (LCFAs) [146,147]. FABP5 also stimulates the transport of FAs to specific intracellular compartments from the cytoplasm to organelles [148]. Furthermore, FABP5 can indirectly interact with membranes, ion channels, receptors, enzymes, or genes, thereby modulating the concentrations of FAs and related molecules and acting as a mediator in various cellular processes [149]. As an example of the physiological contribution of FABP5, it was reported that FABP5 originating from capillary endothelial cells is involved in the uptake of circulating FAs into cardiac and skeletal myocytes to maintain enough levels of ATP production [148]. Another study showed that a high affinity of FABP5 for palmitate is physiologically required to form the major surfactant phospholipid and dipalmitoyl phosphatidylcholine in lung type II alveolar cells [150,151]. In ocular cells, it was shown that knockdown of mRNA of FABP5 in ARPE-19 cells induced various dysregulated FA metabolisms, alteration of cellular lipid composition, and decrease of apolipoprotein B100 levels to maintain cellular homeostasis [152]. Therefore, FABP5 has essential roles in lipid metabolism by coordinating FAs, thereby contributing to several physiological functions including signal transduction, lipid droplet storage, trafficking and membrane synthesis in the endoplasmic reticulum, oxidation in mitochondria or peroxisomes, regulation of the activity of cytosolic and other enzymes, and lipid-mediated transcriptional regulation in the nucleus [153]. Furthermore, since FABP5 is also involved in the regulation of systematic glucose levels, lipid homeostasis, energy metabolism, cell proliferation, and immunological regulations under pathological conditions, FABP5 plays a crucial role in the pathogenesis of various diseases and disorders including metabolism disorders such as obesity, insulin resistance, and type 2 DM [81], skin diseases such as psoriasis [154], neurological diseases such as Alzheimer’s disease [155], and malignant tumors including prostate cancer, breast cancer, and cervical cancer tumors [156], suggesting that FABP5 has great potential in clinical applications. PAX6 is known as the critical transcription factor to be essentially involved in ocular development in vertebrates including the retina [157] and lens [158], as well as various ocular pathogenesis such as aniridia [159,160]. FABP5 is also identified in intraocular tissue, including the lens [161], suggesting that FABP5 may be related to PAX6. In fact, a recent study using the siRNA aniridia cell model strongly suggested that FABP5 expression is regulated by PAX6 [162]. 

Our recent studies showed the presence of ioFABP4 in patients with RVDs and suggested that ioFABP4 may be an additional key target factor for the pathogenesis of RVDs other than ioVEGFA [9,10]. We also found that the FABP family member FABP5 was present in vitreous specimens surgically obtained from patients with RVDs, including PDR (n = 20), RVO (n = 10), and ERM (n = 18), and levels of ioFABP5 were also significantly elevated in patients with RVDs compared with levels in patients with ERM, as is the case of FABP4 [11,12]. However, interestingly, in RVD patients, the elevated levels of ioFABP5 were different from those of ioFABP4: levels of ioFABP5 in patients with RVO and PDR were significantly higher than those in patients with ERM, and elevated levels were more evident in patients with RVO as compared with levels in patients with PDR, but the levels of ioFABP4 in patients with PDR were much higher than levels in patients with RVO and ERM, and there were almost no differences among the groups in levels of ioVEGFA (Table 4) [11]. Furthermore, correlation analyses and multivariable regression analyses indicated that ioFABP5 may be differently involved in the pathogenesis of RVD with ioFABP4 and ioVEGFA (Table 5) [11,12]. Collectively, the observations suggested that levels of ioFABP5 and ioFABP4 might be preferentially influenced by the atherosclerosis-related retinal pathogenesis of RVO and the DM-related retinal pathogenesis of PDR, respectively. In support of this idea, a previous study showed that FABP5 is pivotally involved in the pathogenesis of the early stages of atherosclerosis [72], and FABP5 was also identified within the blood-brain barrier (BBB), which is known to be similar to the BRB [163], and was shown to function in the transport of DHA [164,165]. In fact, in our recent study, Log ioFABP5, but not Log ioFABP4 or Log ioVEGFA, was found to be significantly and negatively correlated with several indices of ocular blood flow, determined by laser speckle flowgraphy (Table 5) [11,12]. As of this writing, the mechanisms underlying these differences between ioFABP4 and ioFABP5 remain to be elucidated, although both FABP4 and FABP5 are expressed in endothelial cells, as well as in adipocytes and macrophages [2], and are secreted into bodily fluids, and both factors are thus involved in the pathophysiological conditions related to several metabolic and cardiovascular diseases [1,68]. However, as a possible mechanism, we speculate that ioFABP5 may solely, or in cooperation with ioVEGFA, be involved in the pathogenesis of RVDs following the inflammatory damage of retinal endothelial cells. In fact, it was recently revealed that FABP5 upregulates the expression of VEGF, a key factor that promotes angiogenesis and metastasis, in prostate cancer [166,167].

## 7. Summary of Current Concepts of the Biological Roles of ioFABP4 and ioFFAs

Intraocularly, the adipocyte related-factor FABP4 (ioFABP4) and its substrates, FFAs (ioFFAs), are present, despite the fact that no adipose tissues are present, and both ioFABP4 and ioFFAs are of intraocular origin and not from peripheral blood circulation. In our recent studies [9,10,11,12], we found that one of the possible producing cells of ioFABP4 was the ocular choroid. The ioFABP4 produced from the ocular choroid is then secreted and distributed into the sensory retina and vitreous cavity via oBRB, thereby playing pivotal roles in retinal physiology and pathogenesis (Figure 4). As one possible scenario, we advocate that the ocular choroid/sensory retinal complex consisting of the ocular choroid and the conventional sensory retina (from RPE to NFL) may help a better understanding of retinal physiology and pathogenesis if the contributions of ioFABP4 and ioFFAs are taken into consideration. However, a recent single-cell RNA sequencing study (GSE137537 and GSE137847) [8] showed that: (1) expression profiles of FABP3 and FABP4 or FABP7 and FABP8 among various retinal cells were similar or different with our immunohistochemistry [12]; and (2) positive retinal expression of FABP5 and FABP12 was not detected in our immunohistochemistry [12]. Therefore, we believe that advances in studies on ioFABP4 and ioFFAs will provide clues for a deeper understanding of intraocular pathophysiology and new therapeutic strategies for vitreoretinal diseases. Since it is also suggested that other FABP family proteins are also involved in some pathophysiological roles in the intraocular environment, additional studies on ioFABPs and ioFFAs will facilitate new research fields to investigate unidentified roles of FABPs, FFAs, and other lipid-related regulatory factors in non-adipose tissues.

## Figures and Tables

**Figure 1 ijms-25-07717-f001:**
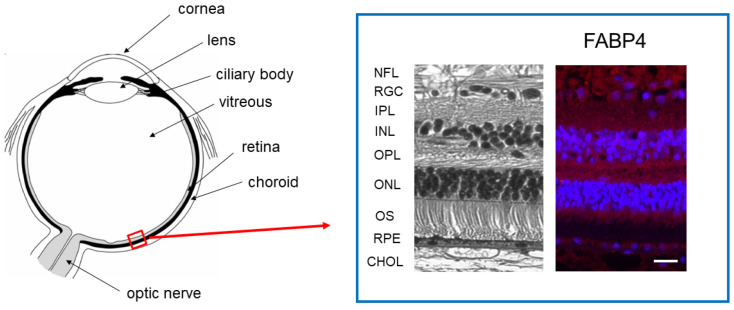
Retinal distribution of fatty acid-binding protein 4 (FABP4) of healthy human retina. Retinal distribution of FABP4 in a healthy human eye was determined by immunohistochemistry using anti-FABP4 antibody (red) and DAPI (blue). Representative hematoxylin and eosin stain and immunolabeling images are recreated using images in our recent study [12]. NFL: nerve fiber layer, RGC: retinal ganglion cell layer, IPL: inner plexiform layer, INL: inner nuclear layer, 0PL: outer plexiform layer, ONL: outer nuclear layer, OS: photoreceptor outer segments, RPE: retinal pigment epithelium, CHOL: ocular choroid. Scale bar; 100 μm.

**Figure 2 ijms-25-07717-f002:**
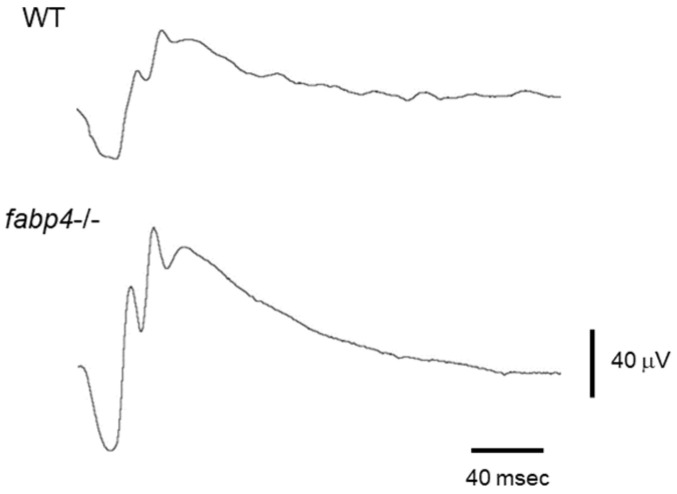
Representative full-field flash electroretinograms of wild-type (WT) and *fabp4*^−/−^ mice. ERG amplitudes (mean ± SEM, n = 5 each): a-wave (WT: 131 mV ± 20.7, *fabp4*^−/−^: 215 mV ± 25.3), b-wave (WT: 250 mV ± 34.1, *fabp4*^−/−^: 371 mV ± 75.9).

**Figure 3 ijms-25-07717-f003:**
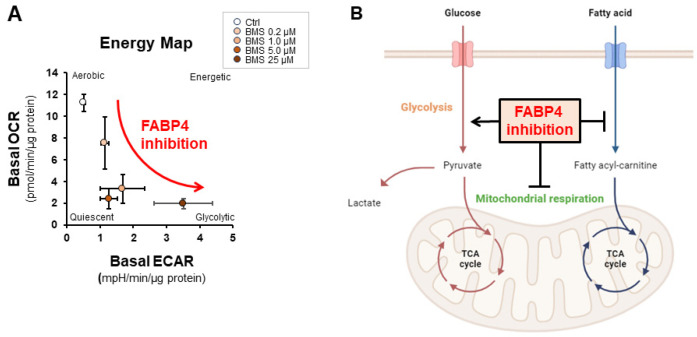
Roles of FABP4 in intraocular metabolism. Panel (**A**): Effects of an FABP4 inhibitor (BMS309430) on cellular metabolism in HOCF cells. An energy map of basal metabolism evaluated by using an extracellular flux analyzer is presented with oxygen consumption rate (OCR) and extracellular acidification rate (ECAR). Reproduced from Ohguro et al. [131]. Panel (**B**): A schematic summary of the metabolic phenotype of FABP4 inhibition in intraocular cells. The illustration was created with BioRender.com. TCA: tricarboxylic acid.

**Figure 4 ijms-25-07717-f004:**
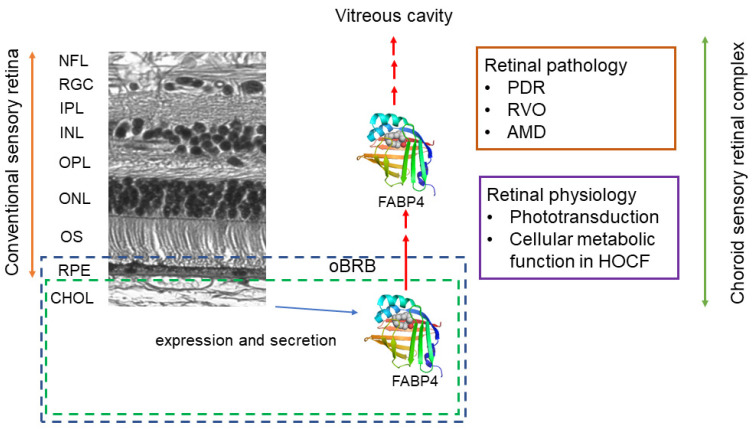
Possible pathophysiological roles of inFABP4. ioFABP4 originates from the ocular choroid and is secreted into the conventional sensory retina and vitreous cavity via oBRB, thereby playing pivotal roles in retinal physiology and pathogenesis. Here, we proposed the so-called ocular choroid/sensory retinal complex when the contributions of ioFABP4 and ioFFAs are taken into consideration in retinal physiology and pathology. NFL: nerve fiber layer, RGC: retinal ganglion cell layer, IPL: inner plexiform layer, INL: inner nuclear layer, OPL: outer plexiform layer, ONL: outer nuclear layer, OS: photoreceptor outer segments, RPE: retinal pigment epithelium, CHOL: ocular choroid. oBRB: outer blood-retinal barrier.

**Table 1 ijms-25-07717-t001:** Localization and pathophysiological aspects of mammalian FABP family.

	Common Name	Aliases	Expressed Tissues and Cells	Biological Roles	Expression Levels of Pathologenic Conditions	Number of Amino Acids/Chromosomal Location
FABP1	Liver FABP (L-FABP)	hepatic FABP, Z protein, heme-binding protein	Liver, alveolar epithelium cells, small intestine, colon, duodenum, kidney	bind FA, bile acids, and exogenous substrates for lipid metabolism and energy homeostasis	liver damage (serum↑)	127/2p11
FABP2	Intestinal FABP (I-FABP)	gut FABP (gFABP)	small intestine, duodenum, colon	transport of exogenous FA to modulate cell growth and proliferation during dietary lipid absorption	colon cancer↑	132/4q28-q31
FABP3	Heart FABP (H-FABP)	O-FABP, mammary-derived growth inhibitor (MDGI)	heart, neurons, glia, kidney, prostate, mammary gland, placenta, ovary, brown adipose tissue	transport of FA and other lipophilic substrates to modulate cell growth and proliferation of muscle and heart	heart failure (serum↑)	133/1p33-p31
FABP4	Adipocyte FABP (A-FABP)	aP2	fat, macrophage, liver, limb, whole brain, placenta	transport of FA in intracellular compartment, export of FA in plasma, and regulation of lipid metabolism	obesity and metabolic syndrome (plasma↑)	132/8q21
FABP5	Epidermal FABP (E-EABP)	keratinocyte-type FABP (K-FABP), psoriasis-associated FABP(PA-FABP)	esophagus, fat, colon, skin, colon, lung, limp node, heart, placenta, neurons, lens, and glia	transport of FA to adjust cellular fatty acid movement of skin and blood circulation	breast and prostate cancers↑	135/8q21.13
FABP6	Ileal FABP (IL-FABP)	ileal lipid-binding protein (ILLBP), intestinal bile acid-binding protein (I-BABP), gastrophin	small intestine	transport of FA to correct absorption and transport of bile acids	colon cancer↑	128/5q23-q35
FABP7	Brain FABP (B-FABP)	brain lipid-binding protein (PLBP), MRG	brain, neurons, glia, skin, salivary gland, fat, retina	transport of FA plays a fundamental role in neurogenesis and astrocyte proliferation	breast cancer, melanoma, renal carcinoma, cystic carcinoma, and invasive glioma↑	132/5q22-q23
FABP8	Myelin FABP (M-FABP)	peripheral myelin protein2 (PMP2)	brain, myelin sheaths of the peripheral nervous system	transport of FA to regulate the structural and functional integrity of myelin	dominant demyelinating Charcot-Marie-Tooth neuropathy↓	132/8q21.3-q22.1
FABP9	Testis FABP (T-FABP)	testis lipid-binding protein (TLBP), PERF15	testis, spleen, fat, brain, endometrium	transport of FA to stabilize spermatogenesis and fertilization	prostate cancer↑	132/8q21.13
FABP12	none	none	retinoblastoma cell, rodent retina, rodent testicular germ cells, rodent cerebral cortex, rodent kidney, rodent epididymis	germinal lipid metabolism	not reported	140/8q21.13

References: FABP1 [13,14], FABP2 [14], FABP3 [15], FABP4 [2,16], FABP5 [17], FABP6 [18], FABP7 [19], FABP8 [19], FABP9 [20], FABP12 [21].

**Table 2 ijms-25-07717-t002:** Correlations of plasma (p-) and intraocular (io-) FFA subspecies.

	ioFFA (µg/mL)	ioC16:0 (µg/mL)	ioC18:0 (µg/mL)	ioC18:1 (µg/mL)	ioC18:2 (µg/mL)	ioC20:4 (µg/mL)
ioFFA (µg/mL)	−0.388					
ioC16:0 (µg/mL)		−0.395				
ioC18:0 (µg/mL)			−0.131			
ioC18:1 (µg/mL)				−0.276		
ioC18:2 (µg/mL)					−0.424	
ioC20:4 (µg/mL)						0.394

Levels of p-FFA and ioFFA were determined by gas-chromatography, as shown in our recent studies [9,10,11,12].

**Table 3 ijms-25-07717-t003:** Retinal distribution of FABP family in human and rodent retinas.

	FABP3		FABP4		FABP5		FABP7		FABP8		FABP12	
	Human	Rodent	Human	Rodent	Human	Rodent	Human	Rodent	Human	Rodent	Human	Rodent
NFL	+	−	+	+	−	−	+	−	±	−	−	−
GCL	+	−	+	+	−	−	+	−	−	−	−	±
IPL	+	−	+	+	−	−	+	−	±	−	−	−
INL	+	−	+	+	−	−	+	−	−	−	−	+
OPL	+	−	+	+	−	−	+	−	−	−	−	−
ONL	+	−	+	+	−	−	+	−	−	−	−	+
OS	±	−	−	−	−	−	±	−	−	−	−	−
RPE	+	−	+	+	−	−	−	−	+	−	−	−

Retinal distribution of FABP isoforms of healthy human retinas and rodent retinas was determined by immunohistochemistry using specific antibodies against FABP isoforms, as shown in our recent study [12]. NFL; nerve fiber layer, RGC; retinal ganglion cell layer, IPL; inner plexiform layer, INL; inner nuclear layer, OPL; outer plexiform layer, ONL; outer nuclear layer, OS; photoreceptor outer segments, RPE; retinal pigment epithelium.

**Table 4 ijms-25-07717-t004:** Correlations of intraocular (io-) FABP4, VEGFA, FFA subspecies, and ocular blood flow levels.

	ERM	PDR	RVO
ioFABP4 (ng/mg protein)	0.30 (0.26–0.35)	1.14 (0.65–3.03) *	0.36 (0.30–0.61) †
ioFABP5 (ng/mg protein)	0.24 (0.11–0.32)	0.84 (0.47–1.14) *	1.38 (0.77–3.29) *^,^†
ioVEGFA (mg/mg protein)	6.8 (5.8–8.4)	166.4 (50.3–295.1) *	12.9 (3.6–35.2)
ioFFA (µg/mL)	1.37 (0.52–4.18)	15.2 (8.3–30.2) *	8.8 (1.9–14.1)
ioC16:0 (µg/mL)	0.65 (0.23–1.68)	4.3 (2.1–8.0) *	2.1 (0.3–3.7)
ioC18:0 (µg/mL)	0.16 (0.02–1.23)	1.9 (0.4–3.2) *	1.00 (0.02–2.25)
ioC18:1 (µg/mL)	0.02 (0.02–0.60)	3.8 (1.4–7.4) *	2.1 (0.7–3.5) *
ioC18:2 (µg/mL)	0.02 (0.02–0.53)	3.3 (1.5–7.1) *	1.9 (0.4–4.0) *
ioC20:4 (µg/mL)	0.05 (0.05–0.05)	1.20 (0.05–2.50) *	0.60 (0.05–1.15)

* *p* < 0.05 vs. ERM; † *p* < 0.05 vs. PDR. Ocular blood flow at the optic disc (OD) at different areas including MBR(A): all areas of the OD, MBR(V): vascular area of the OD, MBR(T): tissue area of the OD, and 4MBR(V)-MBR(T), which were measured as described previously [11]. MBR: mean blur rate, an index for laser speckle flowgraphy, FFA: total fatty acids. C16:0: palmitic acid, C18:0: stearic acid, C18:1: oleic acid, C18:2: linoleic acid, C20:4: arachidonic acid.

**Table 5 ijms-25-07717-t005:** Levels of intraocular (io-) FABP4, VEGFA, and FFA subspecies in vitreous specimens obtained from patients with ERM, PDR, and RVO.

	Log ioFABP4		Log ioFABP5		Log ioVEGFA	
	r	*p*	r	*p*	r	*p*
Log ioFABP4	-	-	0.38	0.008	0.68	<0.001
Log ioFABP5	0.38	0.008	-	-	0.35	0.015
Log ioVEGFA	0.68	<0.001	0.35	0.015	-	-
ioFFA (µg/mL)	0.50	<0.001	0.18	0.262	0.43	<0.001
ioC16:0 (µg/mL)	0.51	<0.001	0.16	0.312	0.42	0.001
ioC18:0 (µg/mL)	0.57	<0.001	0.12	0.453	0.38	0.003
ioC18:1 (µg/mL)	0.49	<0.001	0.22	0.159	0.45	<0.001
ioC18:2 (µg/mL)	0.48	<0.001	0.19	0.231	0.41	0.001
ioC20:4 (µg/mL)	0.43	<0.001	0.17	0.279	0.40	0.002
MBR(A)	−0.43	0.007	−0.57	<0.001	−0.35	0.032
MBR(V)	−0.48	0.003	−0.62	<0.001	−0.35	0.035
MBR(T)	−0.14	0.399	−0.37	0.026	−0.13	0.438
MBR(V)-MBR(T)	−0.53	0.001	−0.62	<0.001	−0.37	0.023
MBR(M)	−0.08	0.639	−0.34	0.037	−0.20	0.231

Levels of intraocular (io-) FABP4, VEGFA, and FFA subspecies were determined in our recent study [11,12]. FFA: total fatty acids. C16:0: palmitic acid, C18:0: stearic acid, C18:1: oleic acid, C18:2: linoleic acid, C20:4: arachidonic acid.

**Table 6 ijms-25-07717-t006:** Correlations of Log ioFABP4, Log ioFABP5, Log ioVEGFA, Log ioFFA subspecies, and ocular blood flow levels.

	Log pFABP4	Log pFABP5	Log pVEGFA	ioFFA (µg/mL)	ioC16:0 (µg/mL)	ioC18:0 (µg/mL)	ioC18:1 (µg/mL)	ioC18:2 (µg/mL)	ioC20:4 (µg/mL)
Log ioFABP4	0.16	-	-	-	-	-	-	-	-
Log ioFABP5	-	n.d.		-	-	-	-	-	-
Log ioVEGFA	-	-	0.0037	-	-	-	-	-	-
ioFFA (µg/mL)	-	-	-	−0.388	-	-	-	-	-
ioC16:0 (µg/mL)	-	-	-	-	−0.395	-	-	-	-
ioC18:0 (µg/mL)	-	-	-	-	-	−0.131	-	-	-
ioC18:1 (µg/mL)	-	-	-	-	-	-	−0.276	-	-
ioC18:2 (µg/mL)	-	-	-	-	-	-	-	−0.424	-
ioC20:4 (µg/mL)	-	-	-	-	-	-	-	-	0.394

Correlations of Log ioFABP4, Log ioFABP5, Log ioVEGFA, Log ioFFA subspecies, and ocular blood flow levels were determined in our recent study [11,12]. Ocular blood flow at the optic disc (OD) at different areas including MBR(A): all areas of the OD, MBR(V): vascular area of the OD, MBR(T): tissue area of the OD, and 4MBR(V)-MBR(T), which were measured as described previously [11]. MBR: mean blur rate, an index for laser speckle flowgraphy, FFA: total fatty acids. C16:0: palmitic acid, C18:0: stearic acid, C18:1: oleic acid, C18:2: linoleic acid, C20:4: arachidonic acid.

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
