# Peer review of "Fatty Acid-Binding Protein 4-Mediated Regulation Is Pivotally Involved in Retinal Pathophysiology: A Review"

_ijms, 2024, doi:10.3390/ijms25147717_

Round 1
Reviewer 1 Report
Comments and Suggestions for Authors
In this present manuscript, Ohguro and colleagues present a comprehensive review of the topic Fatty acid-binding protein (FABPs) 4 in retinal pathophysiology. Overall, the authors have organized the review topics very well, including general information about FABPs in other systems, the distribution of FABPs families comparing humans and rodent retinas, and ultimately the authors have also included relative gene expression (mRNAs) in fabp4 null mice. The manuscript covers updated literature information regarding the review topic and brings new findings in the field.
Few general comments listed below:
1. Line #66, please include the year in the citation “Sellner and Phillips”.
2. I suggest to breakdown the phrase below from lines #92-98: “As shown in Table 2, positive immunoreactivities against FABP3, FABP4 (Fig. 1), FABP7 and FABP8 were detected in healthy human retinas but their localizations were exclusively different among these FABP isoforms, that is, immunolabelings were recognized in all retinal layers for FABP3, all retinal layers except the photoreceptor outer segment (OS) for FABP4, all retinal layers except the OS and RPE 96 for FABP7 and the nerve fiver layer (NFL), inner plexiform layer (IPL) and retinal pigment epithelium (RPE) for FABP8.
3. I suggest rewriting the phrase below from lines #154-159: “However, despite such a strong positive their correlation, additional correlation analyses with various clinical factors and stepwise multiple 155 regression analyses for ioFABP4 and ioVEGFA suggested that both factors were independently regulated. Furthermore, 1) both factors were also negatively correlated with ocular blood flow in the optic nerve head which was more evident with ioFABP4 ioFABP4 levels were not correlated with the presence of vitreous hemorrhaging and its plasma levels (Table 5).
4. I suggest rewriting the phrase (lines #174-177): “Although it had not been elucidated in terms of relationships between FABP4 and VEGFA, previous studies showed that VEGFA and basic fibroblast growth factor 175 (bFGF) can facilitate the expression of FABP4 in endothelial cells, and FABP4 thereby 176 stimulates angiogenesis”
5. Please include more references for the statement (lines #238-239): “In fact, for this purpose, an in vivo mouse model of laser-induced CNV has been extensively used [76].”
6. I suggest including the figure color legends for Figure 3. It is not clear which bars refers to the groups WT, fabp -/- or CNV.
7. Lines #289-295: The authors mention that intraocular FABP4 has been detected in HOCF cells. Have the authors assessed ioFABP4 levels in vivo?
8. The authors have included FABP5 as one of the topics covered by this present manuscript (lines #334-395). FABP5 seems to be regulated by Pax6, as decreased levels of FABP5 are found in PAX6 knockout cells (human-derived) (Katiyar et al., 2022). Pax6 also regulates retinal development (and lens development), and several related diseases. Have the authors addressed that mechanistic relationship?
9. Line #510- correction: In our recent studies, we found that one of origins of ioFABP4 was the ocular choroid.
Comments on the Quality of English Language
The manuscript needs some edits suggested in the review report.
Author Response
Dear Editor,
Thank you very much for the constructive comments concerning our manuscript “Fatty acid-binding protein 4-mediated regulation is pivotally involved in retinal pathophysiology: A review.”. We carefully checked all of the Editor comments and prepared a revised version of our paper that takes these comments into account. The changes are listed below.
Reviewer 1 comments
In this present manuscript, Ohguro and colleagues present a comprehensive review of the topic Fatty acid-binding protein (FABPs) 4 in retinal pathophysiology. Overall, the authors have organized the review topics very well, including general information about FABPs in other systems, the distribution of FABPs families comparing humans and rodent retinas, and ultimately the authors have also included relative gene expression (mRNAs) in fabp4 null mice. The manuscript covers updated literature information regarding the review topic and brings new findings in the field.
Few general comments listed below:
- Line #66, please include the year in the citation “Sellner and Phillips”.
Answer; We sincerely appreciate your excellent comment. In terms of the year in the citation “Sellner and Phillips”, that information is already included in ref# 13: Sellner, P. A.; Phillips, A. R., Phospholipid synthesis by chick retinal microsomes: fatty acid preference and effect of fatty acid binding protein. Lipids 1991, 26, (1), 62-7.
- I suggest to breakdown the phrase below from lines #92-98: “As shown in Table 2, positive immunoreactivities against FABP3, FABP4 (Fig. 1), FABP7 and FABP8 were detected in healthy human retinas but their localizations were exclusively different among these FABP isoforms, that is, immunolabelings were recognized in all retinal layers for FABP3, all retinal layers except the photoreceptor outer segment (OS) for FABP4, all retinal layers except the OS and RPE 96 for FABP7 and the nerve fiver layer (NFL), inner plexiform layer (IPL) and retinal pigment epithelium (RPE) for FABP8.
Answer; We sincerely appreciate your excellent comment. As suggested, that phrase is rewritten to be more easily understood: “As shown in Table 3, immunohistochemistry revealed that positive labeling profiles in the healthy human retinas were exclusively different among FABP isoforms. All retinal layers, those except the photoreceptor outer segment (OS), or those except the OS and RPE were immunolabelled by anti-FABP3 antibody, anti-FABP4 antibody (Fig. 1) or anti-FABP7 antibody, respectively, and anti-FABP8 antibody reacted with the nerve fiver layer (NFL), inner plexiform layer (IPL) and RPE.”.
- I suggest rewriting the phrase below from lines #154-159: “However, despite such a strong positive their correlation, additional correlation analyses with various clinical factors and stepwise multiple 155 regression analyses for ioFABP4 and ioVEGFA suggested that both factors were independently regulated. Furthermore, 1) both factors were also negatively correlated with ocular blood flow in the optic nerve head which was more evident with ioFABP4 ioFABP4 levels were not correlated with the presence of vitreous hemorrhaging and its plasma levels (Table 5).
Answer; We sincerely appreciate your excellent comment. As suggested, this phase and next phrase are rewritten to be more easily understood: “Furthermore, both factors were negatively correlated with ocular blood flow in the optic nerve head and the correlation was stronger for ioFABP4 (Table 5). However, correlation analyses with various clinical factors and stepwise multiple regression analyses suggested that ioFABP4 and ioVEGFA were independently regulated. Levels of ioFABP4 were not correlated with the presence of vitreous hemorrhaging or its plasma levels [10-12]. Collectively, the results suggested that ioFABP4 originates from some intraocularly originating cells, not from peripheral blood circulation, and that ioFABP4 may be an additional key target molecule that is involved in the pathogenesis of PDR by affecting ocular blood circulation.”.
- I suggest rewriting the phrase (lines #174-177): “Although it had not been elucidated in terms of relationships between FABP4 and VEGFA, previous studies showed that VEGFA and basic fibroblast growth factor 175 (bFGF) can facilitate the expression of FABP4 in endothelial cells, and FABP4 thereby 176 stimulates angiogenesis”
Answer; We sincerely appreciate your excellent comment. As suggested, this phase and next phrase are rewritten to be more easily understood: “Regarding relationships between FABP4 and VEGFA, previous studies showed that VEGFA and basic fibroblast growth factor (bFGF) can facilitate the expression of FABP4 in endothelial cells, and FABP4 thereby stimulates angiogenesis [82].”.
- Please include more references for the statement (lines #238-239): “In fact, for this purpose, an in vivo mouse model of laser-induced CNV has been extensively used [76].”
Answer; We sincerely appreciate your excellent comment. As suggested, additional references (PMID: 26779879, PMID: 31270802, PMID: 38007487) are included.
- I suggest including the figure color legends for Figure 3. It is not clear which bars refers to the groups WT, fabp -/- or CNV.
Answer; We sincerely appreciate your excellent comment. As suggested by other expert reviewer, unpublished data should not be included, and thus, Figures 2 and 3 related to laser CNV data were removed.
- Lines #289-295: The authors mention that intraocular FABP4 has been detected in HOCF cells. Have the authors assessed ioFABP4 levels in vivo?
Answer; We sincerely appreciate your excellent comment. In terms of ioFABP4, we detected in vivo ioFABP4 from vitreous specimens surgically obtained from patients with ERM and RVD, but expression of FABP4 in HOCF cell was assessed by only in vitro, but not in vivo. Therefore, this information is included in 1st paragraph of roles of intraocular FABP4 (ioFABP4) in retinal physiology; “To obtain additional insight into the physiological significance of ioFABP4, its intraocular origin was investigated using four representative intraocular tissue-derived cell types including human non-pigmented ciliary epithelium cells, retinoblastoma cells, ARPE19 cells and human ocular choroidal fibroblast (HOCF) cells [124]. Based on the result that among these cells, gene expression of FABP4 was only detected in HOCF cells, we suggested that ocular choroidal tissue is one of possible producing cells of ioFABP4 within the intraocular environment. However, since gene expression in these immortalized cell lines may not reflect in vivo expression of genes in their corresponding cell types in the retina, additional experimental evidences will be required.”.
- The authors have included FABP5 as one of the topics covered by this present manuscript (lines #334-395). FABP5 seems to be regulated by Pax6, as decreased levels of FABP5 are found in PAX6 knockout cells (human-derived) (Katiyar et al., 2022). Pax6 also regulates retinal development (and lens development), and several related diseases. Have the authors addressed that mechanistic relationship?
Answer; We sincerely appreciate your excellent comment. As suggested, those important issues are included: “PAX6 is known as the critical transcription factor to be essentially involved in ocular development in vertebrates including retina [150] and lens [151] as well as various ocular pathogenesis such as aniridia [152, 153]. Since FABP5 is also identified intraocular tissue including lens [154] suggesting FABP5 may be related to Pax6. In fact, a recent study using siRNA aniridia cell model strongly suggested that FABP5 expression is regulated by PAX6 [155].”.
- Line #510- correction: In our recent studies, we found that one of origins of ioFABP4 was the ocular choroid.
Answer; We sincerely appreciate your excellent comment. As pointed out, this sentence is changed to “we found that one of possible producing cells of ioFABP4 was the ocular choroid.” to avoid misunderstanding.
- Comments on the Quality of English Language: The manuscript needs some edits suggested in the review report.
Answer; We sincerely appreciate your excellent comment. Quality of English was carefully checked by a native English speaking Scientist.
Reviewer 2 comments
This is review of fatty acid binding proteins and the particular importance of fatty acid-binding protein 4 (fabp4) in retinal physiology and disease. It is generally well written and will be useful to those interested in retinal lipid metabolism and retinal diseases such as age related macular degeneration and diabetic retinopathy.
- While the authors rely on their own published results, they cite the work of many groups even when those papers report different results.
Answer; We sincerely appreciate your excellent comment. As suggested, our unpublished works were deleted and additional other groups works (PAX6 related to FABP5, scRNA sequencing and other references suggested by other expert reviewers) were included.
- Ohguro and colleagues cite their own unpublished work in two places in this review, and that is inappropriate, since the readers will not be able to access them.
Answer; We sincerely appreciate your excellent comment. I agree with this comment and thus inappropriate our unpublished works (laser CNV experiment using fabp4 deficiency mice and effects of FA on RPE) were removed.
- The current paper has a section on the role of free fatty acids in retinal pathogenesis, and the authors should consider moving this section early in the paper, before discussing fatty acid binding proteins.
Answer; We sincerely appreciate your excellent comment. As your excellent suggestion, section of FA is moved in advance to FABP related issues.
- The authors draw a major conclusion, that intraocular fabp4 derives solely from the choroid, from gene expression data obtained in immortalized cell lines. Since pattern of gene expression in these cells does not accurately correspond to the gene expression of photoreceptors, RPE and choroidal endothelial cells in vivo, they should moderate their claims or substantiate them with scRNAseq data. These data should be available in publicly available databases such as GEO Accession no. GSE137847.
Answer; We sincerely appreciate your excellent comment. I totally agree with your critical comment that pattern of gene expression in these cells does not accurately correspond to the gene expression of photoreceptors, RPE and choroidal endothelial cells in vivo. Therefore, using this information, last paragraph of summary is rewritten: “Intraocularly, the adipocyte related-factor FABP4 (ioFABP4) and its substrates, FFAs (ioFFAs), are present despite the fact that no adiposed tissues are present, and both ioFABP4 and ioFFAs are of intraocular origin and not from peripheral blood circulation. In our recent studies [9-12], we found that one of possible producing cells of ioFABP4 was the ocular choroid. The ioFABP4 originating from the ocular choroid is then secreted and distributed into the sensory retina and vitreous cavity via oBRB, thereby playing pivotal roles in retinal physiology and pathogenesis (Fig. 6). As one of possible scenario, we advocate that the ocular choroid/sensory retinal complex consisting from the ocular choroid and the conventional sensory retina (from RPE to NFL) may help a better understanding of retinal physiology and pathogenesis if the contributions of ioFABP4 and ioFFAs are taken into consideration. However, a recent single cell RNA sequencing study (GSE137537 and GSE137847) [8] showed that 1) expression profiles of FABP3 and FABP4 or FABP7 and FABP8 among various retinal cells were similar or different with our immunohistochemistry [12] and 2) positive retinal expression of FABP5 and FABP12 which were not detected in our immunohistochemistry [12]. Therefore, we believe that advances in studies on ioFABP4 and ioFFAs will provide clues for a deeper understanding of intraocular pathophysiology and new therapeutic strategy for vitreoretinal diseases. Since it is also suggested that other FABP family proteins are also involved in some pathophysiological roles in the intraocular environment, additional Studies on ioFABPs and ioFFAs will facilitate new research field to investigate unidentified roles of FABPs, FFAs and other lipid-related regulatory factors in non-adipose tissues.”.
Specific comments:
- 2: Change the headings in Table I so that the final heading fits better.
Answer; We sincerely appreciate your excellent comment. A suggested, heading of Table 1 is changed: “Localization and pathophysiological aspects of mammalian FABP family”.
- Lines 104-106: The figure 1 legend should specify that the micrographs are of a human retina, since the previous paragraph ends with mention of rodent retinas.
Answer; We sincerely appreciate your excellent comment. As pointed out, “Fig. 1” in line 102 was wrong, and thus this is deleted and title of Figure 1 is changed to “Retinal distribution of fatty acid-binding protein 4 (FABP4) of healthy human retina”.
- Lines 107ff: Arrange Table 2 to fit on one page.
Answer; We sincerely appreciate your excellent comment. As suggested, Table 2 is suitably arranged.
- Lines 127ff: The section on proliferative diabetic retinopathy is wordy and awkward in spots. It should be re-written.
Answer; We sincerely appreciate your excellent comment. Quality of English was carefully checked by a native English speaking Scientist.
- Line 128: “Thread vision loss” makes no sense. What word is intended?
Answer; We sincerely appreciate your excellent comment. As pointed out, that is changed: “to that can induce vision loss”.
- Table 4: Does “Log” indicate logarithm base 10? If so, say so.
Answer; We sincerely appreciate your excellent comment. In terms of this issue, “Log” indicates logarithm base e (natural logarithm).
- 8 and Fig. 2, Laser injuries can be quite variable. The appearance of one laser induced injury in one mouse is not sufficient to argue that fabp4 deletion suppresses CNV in this model. The results of multiple sports in multiple mice should be tabulated.
Answer; We sincerely appreciate your excellent comment. As suggested above, results of our laser-CNV models using FABP4 deficiency mice are removed, and therefore, corresponding chapter is rewritten: “AMD has been recognized as the most common cause of permanent visual deterioration in the aged population worldwide, resulting in serious problems in public health [111, 112]. As the main pathogenic process of AMD, choroidal neovascularization (CNV) is known to lead to severe vision loss in neovascular AMD due to increased expression of ioVEGF [113, 114], and anti-VEGF therapies have been successfully used to treat neovascular AMD [115]. However, such anti-VEGF therapies have several limitations including the requirement of repeat therapy, development of drug resistance, and large costs for patients, but, unfortunately, there is currently no alternative and satisfactory treatment for neovascular AMD [116]. Therefore, other options for cost-effective, less invasive, and more durable therapy for CNV in AMD patients are required. For this purpose, a therapeutic target molecule that substitutes for VEGF will be needed. As of this writing, there is no evidence of a contribution to FABP4 on AMD pathogenesis. However, previous studies have shown a significant contribution of macrophages to CNV formation in animal models and AMD patients [117-119] and levels of cytokine production and proinflammatory mediators, including TNFα and COX2, were reduced in macrophages isolated from fabp4-deficient (fabp4-/-) mice [120], suggesting that FABP4 may be involved in the pathogenesis of AMD. To support this possibility, a previous study using an oxygen-induced retinopathy (OIR) model [121] in wild-type (WT) and fabp4-/- mice showed that OIR induction in fabp4-/- mice caused a significant decrease in neovessel formation and a significant improvement in physiological revascularization of avascular retinal tissues [122]. Furthermore, it has been shown that FABP4 is an inducible factor for angiogenesis and vascular smooth muscle cell proliferation and migration, and FABP4 has therefore been established as a reliable predictive biomarker for cardiovascular disease in specific at-risk groups [123]. Collectively, the observations suggested that ioFABP4 may be a new risk factor as well as a possible therapeutic target for CNV in AMD.”.
- 10 and Fig. 4, ERG responses are variable, Single ERG tracings at one light intensity to not confirm a difference between wild type and fabp4-/- mice.
Answer; We sincerely appreciate your excellent comment. In terms of this issue, we performed ERG analysis using 5 mice of WT and fabp4-/- mice, and therefore, ERG amplitude od a and b waves are included in the figure legend. In addition, corresponding phrase is changed to remove significantly: “In a recent study, to elucidate the unknown physiological roles of ioFABP4, electroretinograms (ERGs) of WT and fabp4-/- mice were analyzed and it was found that both ERG a- and b-waves were larger in fabp4-/- mice than in WT mice (Fig. 2)”.
- Line 292: In a review paper, it is inappropriate to cite a paper that is not widely available. Just state “unpublished data” or wait until the paper is published or publish a preprint.
Answer; We sincerely appreciate your excellent comment. In terms of this study, this is already published and can be found in PUBMED (PMID: 38927820), and this reference is cited.
- Gene expression in these immortalized cell lines may not reflect the expression of genes in their corresponding cell types in the retina. This is known to be true for undifferentiated ARPE-19 cells grown on plastic plates. The authors can refer to publicly accessible gene expression databases for the retina to corroborate or reject their findings in cultured cells.
Answer; We sincerely appreciate your excellent comment. I totally agree that Gene expression in these immortalized cell lines may not reflect the expression of genes in their corresponding cell types in the retina. However, as stated in comment #9, this study demonstrated not only mRNA expression but also functional assay of cellular metabolic analysis using a Seahorse Bioanalyzer indicating that HOCF cells were indeed affected by a specific inhibitor of FABP4. In addition, in this study (PMID: 38927820), several reviewers also claimed to similar points and finally agreed our proper response. In addition, we are not saying that HOCF cells are truly producing cells of ioFABP4, but only suggested that HOCF cells are possible producing cell of FABP4. Therefore, to avoid misunderstanding, corresponding phases are changed: “Based on the result that among these cells, gene expression of FABP4 was only detected in HOCF cells, we suggested that ocular choroidal tissue is the one of possible producing cells of ioFABP4 within the intraocular environment. However, since gene expression in these immortalized cell lines may not reflect in vivo expression of genes in their corresponding cell types in the retina, additional experimental evidences will be required.”.
- Line 453ff (Table 5): This table can be condensed to take up less space by reporting only the correlations with a numeric value and omitting the minus symbols.
Answer; We sincerely appreciate your excellent comment. As suggested, Table 5 is fixed to omit the minus symbols.
- Line 482: In it is inappropriate to cite work that is submitted for publication. Either publish it as a preprint, and cite it, or state “unpublished results”.
Answer; We sincerely appreciate your excellent comment. As suggested, this observation is removed.
Reviewer 3 comments
The manuscript entitled “Fatty acid-binding protein 4-mediated regulation is pivotally 2 involved in retinal pathophysiology: A review” by Ohguro and co-workers, reviews recent findings on the biologic role of FABP4 in normal and pathological conditions, with a specific focus on retinal diseases. This review includes studies carried out by the authors and compiles evidence to support interesting insights on retinal physiopathology that may contribute to develop new therapeutic strategies. The manuscript is generally well organized and written, although some concerns must be addressed.
Specific comments.
- Figure 1. Is this a previously published figure? If so, please provide the reference. Is it an immunolocalization? If these are not published results, identify the specific signals and provide sufficienn experimental information (antibodies, controls, etc.)
Answer; We sincerely appreciate your excellent comment. In terms of retinal images of HE staining and immunohistochemistry, a previous figure was re-created and thus those informations are included: “Retinal distribution of FABP4 of healthy human eye was determined by immunohistochemistry using anti-FABP4 antibody (red) and DAPI (blue). Representative hematoxylin and eosin stain and immunolabeling images are recreated using images in our recent study [12]. NFL: nerve fiver layer, RGC: retinal ganglion cell layer, IPL: inner plexiform layer, INL: inner nuclear layer, 0PL: outer plexiform layer, ONL: outer nuclear layer, OS: photoreceptor outer segments, RPE: retinal pigment epithelium, CHOL: ocular choroid. Scale bar; 100 mm.”.
- Tables and figure 2 correspond to published studies. The corresponding references must be provided.
Answer; We sincerely appreciate your excellent comment. As suggested other expert reviewer, figures 2 and 3 does not correspond to published data, and thus those were removed. Tables 2-5 correspond to published studies and therefore, corresponding references are included.

Reviewer 2 Report
Comments and Suggestions for Authors
This is review of fatty acid binding proteins and the particular importance of fatty acid-binding protein 4 (fabp4) in retinal physiology and disease. It is generally well written and will be useful to those interested in retinal lipid metabolism and retinal diseases such as age related macular degeneration and diabetic retinopathy. While the authors rely on their own published results, they cite the work of many groups even when those papers report different results. Ohguro and colleagues cite their own unpublished work in two places in this review, and that is inappropriate, since the readers will not be able to access them. The current paper has a section on the role of free fatty acids in retinal pathogenesis, and the authors should consider moving this section early in the paper, before discussing fatty acid binding proteins. The authors draw a major conclusion, that intraocular fabp4 derives solely from the choroid, from gene expression data obtained in immortalized cell lines. Since pattern of gene expression in these cells does not accurately correspond to the gene expression of photoreceptors, RPE and choroidal endothelial cells in vivo, they should moderate their claims or substantiate them with scRNAseq data. These data should be available in publicly available databases such as GEO Accession no. GSE137847.
Specific comments:
p. 2: Change the headings in Table I so that the final heading fits better.
Lines 104-106: The figure 1 legend should specify that the micrographs are of a human retina, since the previous paragraph ends with mention of rodent retinas.
Lines 107ff: Arrange Table 2 to fit on one page.
Lines 127ff: The section on proliferative diabetic retinopathy is wordy and awkward in spots. It should be re-written.
Line 128: “Thread vision loss” makes no sense. What word is intended?
Table 4: Does “Log” indicate logarithm base 10? If so, say so.
P. 8 and Fig. 2, Laser injuries can be quite variable. The appearance of one laser induced injury in one mouse is not sufficient to argue that fabp4 deletion suppresses CNV in this model. The results of multiple sports in multiple mice should be tabulated.
p. 10 and Fig. 4, ERG responses are variable, Single ERG tracings at one light intensity to not confirm a difference between wild type and fabp4-/- mice.
Line 292: In a review paper, it is inappropriate to cite a paper that is not widely available. Just state “unpublished data” or wait until the paper is published or publish a preprint.
Gene expression in these immortalized cell lines may not reflect the expression of genes in their corresponding cell types in the retina. This is known to be true for undifferentiated ARPE-19 cells grown on plastic plates. The authors can refer to publicly accessible gene expression databases for the retina to corroborate or reject their findings in cultured cells.
Line 453ff (Table 5): This table can be condensed to take up less space by reporting only the correlations with a numeric value and omitting the minus symbols.
Line 482: In it is inappropriate to cite work that is submitted for publication. Either publish it as a preprint, and cite it, or state “unpublished results”.
Comments on the Quality of English Language
This is review of fatty acid binding proteins and the particular importance of fatty acid-binding protein 4 (fabp4) in retinal physiology and disease. It is generally well written and will be useful to those interested in retinal lipid metabolism and retinal diseases such as age related macular degeneration and diabetic retinopathy. While the authors rely on their own published results, they cite the work of many groups even when those papers report different results. Ohguro and colleagues cite their own unpublished work in two places in this review, and that is inappropriate, since the readers will not be able to access them. The current paper has a section on the role of free fatty acids in retinal pathogenesis, and the authors should consider moving this section early in the paper, before discussing fatty acid binding proteins. The authors draw a major conclusion, that intraocular fabp4 derives solely from the choroid, from gene expression data obtained in immortalized cell lines. Since pattern of gene expression in these cells does not accurately correspond to the gene expression of photoreceptors, RPE and choroidal endothelial cells in vivo, they should moderate their claims or substantiate them with scRNAseq data. These data should be available in publicly available databases such as GEO Accession no. GSE137847.
Specific comments:
p. 2: Change the headings in Table I so that the final heading fits better.
Lines 104-106: The figure 1 legend should specify that the micrographs are of a human retina, since the previous paragraph ends with mention of rodent retinas.
Lines 107ff: Arrange Table 2 to fit on one page.
Lines 127ff: The section on proliferative diabetic retinopathy is wordy and awkward in spots. It should be re-written.
Line 128: “Thread vision loss” makes no sense. What word is intended?
Table 4: Does “Log” indicate logarithm base 10? If so, say so.
P. 8 and Fig. 2, Laser injuries can be quite variable. The appearance of one laser induced injury in one mouse is not sufficient to argue that fabp4 deletion suppresses CNV in this model. The results of multiple sports in multiple mice should be tabulated.
p. 10 and Fig. 4, ERG responses are variable, Single ERG tracings at one light intensity to not confirm a difference between wild type and fabp4-/- mice.
Line 292: In a review paper, it is inappropriate to cite a paper that is not widely available. Just state “unpublished data” or wait until the paper is published or publish a preprint.
Gene expression in these immortalized cell lines may not reflect the expression of genes in their corresponding cell types in the retina. This is known to be true for undifferentiated ARPE-19 cells grown on plastic plates. The authors can refer to publicly accessible gene expression databases for the retina to corroborate or reject their findings in cultured cells.
Line 453ff (Table 5): This table can be condensed to take up less space by reporting only the correlations with a numeric value and omitting the minus symbols.
Line 482: In it is inappropriate to cite work that is submitted for publication. Either publish it as a preprint, and cite it, or state “unpublished results”.
Author Response

(The authors gave the same response as above.)

Reviewer 3 Report
Comments and Suggestions for Authors
The manuscript entitled “Fatty acid-binding protein 4-mediated regulation is pivotally 2 involved in retinal pathophysiology: A review” by Ohguro and co-workers, reviews recent findings on the biologic role of FABP4 in normal and pathological conditions, with a specific focus on retinal diseases. This review includes studies carried out by the authors and compiles evidence to support interesting insights on retinal physiopathology that may contribute to develop new therapeutic strategies. The manuscript is generally well organized and written, although some concerns must be addressed.
Specific comments.
1.- Figure 1. Is this a previously published figure? If so, please provide the reference. Is it an immunolocalization? If these are not published results, identify the specific signals and provide sufficienn experimental information (antibodies, controls, etc.)
2.- Tables and figure 2 correspond to published studies. The corresponding references must be provided.
Author Response

(The authors gave the same response as above.)

Round 2
Reviewer 2 Report
Comments and Suggestions for Authors
The authors have responded well (and quickly) to the concerns raised in the original review. In line 276, please change " to that can induce vision loss" to "that can induce vision loss".
Comments on the Quality of English LanguageThe authors have responded well to the concerns raised in the original review. In line 276, please remove the word "to" so that the sentence reads: "that
can induce vision loss ..."